image processing/bioengineering/geophysics

gold, nanoparticle, soil, x-ray, contrast, stability

**Author for correspondence:**
Tiina Roose
e-mail: t.roose@soton.ac.uk

# Stabilizing gold nanoparticles for use in X-ray computed tomography imaging of soil systems

Callum P. Scotson[1], Maria Munoz-Hernando[2], Simon J. Duncan[1], Siul A. Ruiz[1], Samuel D. Keyes[1], Arjen van Veelen[1], Iain E. Dunlop[2] and Tiina Roose[1]

[1]Bioengineering Sciences Research Group, Mechanical Engineering, Faculty of Engineering and Physical Sciences, University of Southampton, Southampton, UK
[2]Department of Materials, Faculty of Engineering, Imperial College London, London, UK

TR, 0000-0001-8710-1063

This investigation establishes a system of gold nanoparticles that show good colloidal stability as an X-ray computed tomography (XCT) contrast agent under soil conditions. Gold nanoparticles offer numerous beneficial traits for experiments in biology including: comparatively minimal phytotoxicity, X-ray attenuation of the material and the capacity for functionalization. However, soil salinity, acidity and surface charges can induce aggregation and destabilize gold nanoparticles, hence in biomedical applications polymer coatings are commonly applied to gold nanoparticles to enhance stability in the *in vivo* environment. Here we first demonstrate non-coated nanoparticles aggregate in soil-water solutions. We then show coating with a polyethylene glycol (PEG) layer prevents this aggregation. To demonstrate this, PEG-coated nanoparticles were drawn through flow columns containing soil and were shown to be stable; this is in contrast with control experiments using silica and alumina-packed columns. We further determined that a suspension of coated gold nanoparticles which fully saturated soil maintained stability over at least 5 days. Finally, we used time resolved XCT imaging and image based models to approximate nanoparticle diffusion as similar to that of other typical plant nutrients diffusing in water. Together, these results establish the PEGylated gold nanoparticles as potential contrast agents for XCT imaging in soil.

# 1 Introduction

X-ray computed tomography (XCT) is a mature technology for biomedical and clinical imaging [1], and is increasingly used for non-destructive 3D imaging of soil and plant root systems [2,3]. However, the ability to distinguish easily between soil and plant roots remains a challenge due to overlapping X-ray attenuation of soil pore water and root material [2], in which the resulting poor contrast can complicate image segmentation. Contrast issues also limit sample sizes—the higher X-ray energies required to penetrate larger samples are proportionally less sensitive to attenuation. As a result, the maximum attainable contrast-to-noise ratio (CNR) generally decreases as sample diameter increases [4].

When similar issues occur within biomedical soft tissue imaging, it is common practice to use solutions or suspensions containing radiopaque elements as contrast agents [5]. A recently introduced class of contrast media is gold nanoparticles (AuNPs), which have become the focus of much biomedical research interest. This is due to the numerous beneficial traits they present. One such notable trait is the capacity for functionalization [6,7]. Functionalization is the addition of a functional group or protein structure either directly onto the nanoparticle or onto a coating which envelopes the nanoparticle [7,8]. Functionalization will often be applied to induce accumulation upon specific materials or biological tissues. This is as opposed to tracer contrast agents which are used to trace liquid flows where the intention is that the contrast media will remain in solution or suspension [5]. There are numerous applications of functionalized nanoparticles as contrast agents in biomedical research [7–9]. For example, Sun *et al.* [10] used a functional heparin coating on gold nanoparticles to induce accumulation on murine livers and produced much improved contrast against un-treated tissue.

An additional beneficial trait of AuNPs exploited in biomedical applications is the high density and atomic number which cause AuNPs to be X-ray attenuating—possessing considerably greater attenuation than iodinated contrast media at equivalent concentrations [5]. The contrast provided by AuNPs has often been observed to be up to 2.7 times greater than iodinated equivalents [10,11]. Therefore, lower concentrations of AuNPs can achieve comparable contrast to iodinated contrast media.

In contrast to these advantageous traits however, AuNPs have previously been observed to become unstable and undergo aggregation when exposed to strong surface charges, salinity and acidity [12–14]. These are all common conditions in biological systems. In biomedical research, the application of a Polyethylene Glycol (PEG) coating to the surface of AuNPs has previously been found to overcome the destabilizing effects of saline and/or acidic conditions [7,15]. Therefore, such coatings ensure stability of the nanoparticles in suspension. The PEG coating also replaces the need for stabilizing agents in the solvent, such as citrate solution, which cannot be maintained in most biological systems.

While the utility of gold nanoparticles is frequently explored in biomedical applications and research, use of gold nanoparticles as an XCT contrast agent has not been attempted for plant roots in soil. AuNPs have seen limited use as a radiographic contrast agent in plants and only in non-soil-borne tissues. However, they have presented promising results in the few studies in which AuNPs were applied to *ex vivo* plant material. For example, Ahn *et al.* [16] submerged dried rice leaf sheaths in an AuNP suspension. The xylem vessels of the leaf sheaths then filled with the AuNP suspension through capillary action, and resulted in the AuNP suspension significantly improving radiographic contrast of xylem structures. However, the use of dried plant tissue outside of soil is unrepresentative of the *in vivo* and *in situ* processes. Furthermore, the domain in which XCT has demonstrated great potential is for imaging roots in soil.

Further to those traits exploited in biomedical applications, AuNPs offer a number of favourable traits that particularly relate to use in soil and plant root systems. In particular, AuNPs present comparatively less phytotoxicity [17,18]. For example, when tobacco (*Nicotiana xanthi* L.) seedlings were exposed to AuNP suspensions, the nanoparticles only appeared to induce visually observable leaf damage after 14 days of exposure [18]. By comparison, Keyes *et al.* [19] exposed winter pea (*Pisum sativum* L. cv. frisson) to iodinated contrast media (Gastrografin) and observed severe tissue damage after 1 day. It is noted that different species may tolerate such contrast agents to a different extent, however the scarcity of literature on the subject makes direct comparisons difficult. Additionally, Sabo-Attwood *et al.* [18] only exposed juvenile plants to AuNPs, which should have a lower capacity to tolerate nanoparticles as the particles could only be dispersed over a relatively smaller volume of tissue. Sabo-Attwood *et al.* [18] also highlighted that the range of available AuNP particle sizes enables selective uptake by organic tissues. Smaller AuNPs (3.5 nm) were observed to have reached the leaves whereas the larger size of nanoparticles used (18 nm) were not detected within the plant vasculature, but instead remained agglomerated on the surface of the roots.

At present, however, the full potential of AuNPs as a contrast agent for use in soil systems has not been realized, because they are susceptible to aggregation or degradation and do not remain in

**Table 1.** Information on the three field soils used in this investigation including: soil type and geographical location of origin.

| soil name | soil type | geographical origin |
| --- | --- | --- |
| Bangor | sand-textured Eutric Cambisol | Abergwyngregyn, North Wales, UK (53.014′ N, −4.001′ W) |
| Dundee | sandy-loam Dystric Cambisol | South Bullionfield, James Hutton Institute, Dundee, UK (56.273′ N, −3.104′ W) |
| Nottingham | sandy-loam Eutric Cambisol | University of Nottingham farm, Bunny, Nottinghamshire, UK (52.860′ N, −1.141′ W) |

suspension, i.e. they destabilize [14,20,21]. This in turn compromises the transport of the nanoparticles through soil pore spaces and prevents them from targeting tissues for functionalization. AuNPs in soil are not only exposed to salinity and acidity, as in biomedical applications, but also the destabilizing effects of soil particulate surface charges. Furthermore, the pH and salinity conditions in soil-water solution or plant tissue fluids are likely more heterogeneous than mammalian fluids. For example, in a healthy human adult it is unusual for blood pH to vary from the baseline of approximately 7.4 by more than 0.1 [22], while the pH of soil solutions from UK soils range between approximately 4 and 8 [23] with additional variation present in intra-substrate niches. For AuNPs to be successfully transferred from current biomedical applications to plant root and soil systems, stability of the nanoparticles must be achieved in these complex biogeochemical environments. It is therefore important to establish whether PEG coating procedures, used in biomedical applications to overcome destabilizing effects, are sufficient for use in soil systems.

The aim of this investigation is to ensure stability of AuNPs under different soil conditions. This includes the use of a PEG polymer coating to mitigate destabilizing properties of the soil environment.

We hypothesized that non-coated gold nanoparticles (NC-AuNP) would become destabilized if exposed to the strong particle surface charges, salinity or acidity, which often characterize soil systems. Further, we hypothesized that application of a PEG coating to gold nanoparticles (PEG-AuNP) would mitigate against these destabilizing effects and keep the nanoparticles within a stable suspension. In order to assess the effects of PEG coating on AuNP stability in soil, the following procedures were undertaken. First, the stability of NC-AuNPs was assessed using a selection of soil-water solutions, with the degree of aggregation quantified using ultraviolet–visible spectroscopy (UV–Vis). Based on the results of this stability test, the AuNPs underwent a PEG coating procedure and were tested for stability in soil-water solutions a second time. Following stabilization in soil-water, the PEG-AuNP suspensions were assessed for their stability when drawn through flow columns of porous media. The suspensions were first assessed in columns containing alumina and silica particles, which possess strong surface charges, to characterize AuNP behaviour when destabilized by particles in a 'soil-analogue' porous medium. Scanning electron microscopy (SEM) was used to further characterize AuNP destabilization in the columns, and a mathematical model was used to estimate a lower bound of the buffer power for AuNPs in silica and alumina. The stability of the PEG-AuNP suspensions was trialled in the soils using flow columns containing the soils and by fully saturating the soils with a concentrated suspension of the PEG-AuNP. Finally, the utility of PEG-AuNPs as tracer contrast agents was assessed by estimating the diffusion rate of AuNPs in soil water, using a combination of time resolved XCT imaging and mathematical modelling.

# 2. Material and methods

## 2.1. Preparation of soil-water solutions

Salinity and acidity are hypothesized to induce aggregation in NC-AuNPs, while coating with a PEG polymer is expected to mitigate this effect. To quantify the influence of real soil-water acidity and salinity on NC-AuNP and PEG-AuNP stability, soil-water solutions were first prepared. These were extracted by centrifugation from samples of three soils excavated from field locations across the UK. Those three field locations were in the following regions: Bangor, Dundee and Nottingham and the three soils are referred to by their geographical origin hence forth (detailed information about the three soils is given in table 1).

For the preparation of the soil-water solutions, each soil was sequentially sieved using British Standard geotechnical sieves (Glenammer Engineering, Ayr, UK) of 2 mm and 1.18 mm aperture size. From the less than 1.18 mm fraction of each soil type, a homogenized 10 g was placed into a 50 ml centrifuge tube, 15 ml of UHQ purified water was added and the tube was shaken vigorously for 1 min. All tubes were then agitated on laboratory rollers for 48 h, allowing the soil water solution to equilibrate.

The tubes underwent centrifugation using a Sorvall Legend XTR (Sorvall, UK) at 8000 revolutions per minute (RPM) for 30 min at 20°C in order to separate the solids from the soil solution. After centrifugation, the supernatant was removed from each tube using a 10 ml syringe in series with a 0.45 µm Millex HA syringe filter tip. This step removed soil particles that may have re-suspended following centrifugation. The resulting soil-water solutions contained many of the soil-borne acids and salts. Therefore, these solutions were used to test the stability of the gold nanoparticles under acid and salt conditions representative of those expected in soil environments. This protocol was executed twice for each of the three soil types: once when stored at room temperature and another where they were kept in a 3°C fridge. This was to ensure changes of acidity with temperature could be considered. Ionization can increase with temperature thus more $H^+$ ions will become active and the solution will become more acidic.

## 2.2. pH of assay of soil-water solutions

The pH was determined for the three soil-water solutions. The pH test panel comprised both sets of soil-water solutions for each of the three soils: those stored at room temperature and those stored at 3°C. Measurement of pH was conducted 24 h after preparation using an Accumet AB150 pH testing kit (Fisher Scientific, UK). A three point calibration using pre-prepared buffer solutions at pH 4.01, 7.00 and 10.00 was used to ensure accuracy of the pH measurements.

## 2.3. Characterization of soil-water solution composition

In order to characterize the composition of the soil-water solutions, high resolution Inductively Coupled Plasma Mass Spectrometry (HR-ICP-MS) was conducted using a Thermo Fisher Scientific ELEMENT 2XR (Thermo Fisher Scientific, UK) at the National Oceanography Centre, Southampton. A sub-sample of each of the three soil-water solutions was prepared for ICP-MS using a multi-step digestion procedure. The initial step in the digestion procedure was the addition of 2.5 ml aqua regia to 1 ml of the soil-water solution to digest and oxidize the potential colloids and the majority of the dissolved organic matter. This solution was left on heat plates in a fume cupboard until the aqua regia had evaporated. Following this, 6 M hydrochloric acid (HCl) was applied to the soil-water solutions to digest any remaining organic matter and once again left on heat plates in a fume cupboard for the HCl to evaporate. The final stage of the sample preparation was the addition of 3% nitric acid ($HNO_3$), which contained beryllium, indium and rhenium as internal standards. These were used as standards since they were unlikely to be found within the samples, had few spectral inferences with analytes of interest and would be chemically stable in these solutions.

## 2.4. Assessment of AuNP stability using UV–Vis

Ultraviolet–visible spectroscopy (UV–Vis) is often used for determining gold nanoparticle size [13,24–26], and to indicate the prevalence of aggregation or degradation of gold nanoparticles as a result of instability. UV–Vis, using a Shimadzu UV-1800 spectrometer (Shimadzu, Japan), was used to assess stability of AuNPs in the soil-water solutions and to inform whether it was necessary to coat the AuNPs in order to enhance stability and prevent aggregation.

For the UV–Vis analysis, 100 nm AuNPs (Sigma Aldrich, UK) at the manufacturer-supplied concentration of $3.84 \times 10^{-2}$ mg Au ml$^{-1}$ were added to aliquots of each of the three soil solutions. AuNPs of 100 nm size were chosen for two reasons. Firstly, were plants to be included within experimental set-ups of future work, 100 nm AuNPs are too large to be taken up through pores in plant root cell walls [27]. Secondly, the polymer coating procedure, described in §2.5 Polymer Coating Process, required several rounds of centrifugation. The ease of performing centrifugation is considerably improved for larger particle sizes as they possess a greater mass and thus require a lower RPM for successful centrifugation. Time-resolved UV–Vis analysis was undertaken on the following samples for NC-AuNP and PEG-AuNP suspensions: soil solution mixed with AuNPs, soil

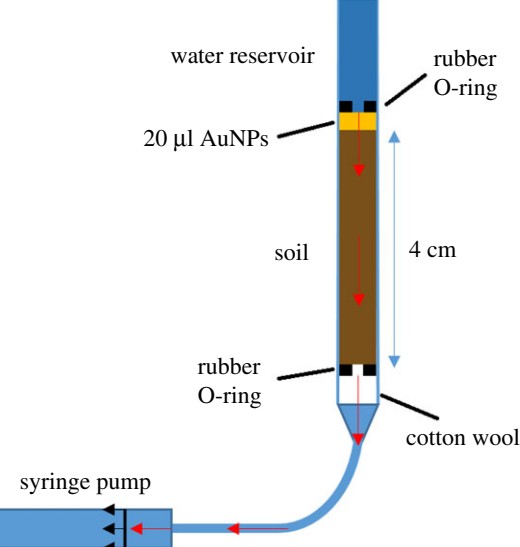

**Figure 1.** Diagram of the constructed experimental system to be used for both soil-analogue and soil columns. Red arrows indicate the movement of water through the saturated system.

solution, UHQ water, and UHQ water mixed with AuNPs. The total volume of each aliquot was 4 ml, at a ratio of 1 ml AuNP to 3 ml of either soil solution or UHQ water.

UV–Vis analysis was also used on extractions of soil-water solutions from the soil columns to assess PEG-AuNP stability. This was applied to the imaged soil columns post-imaging in addition to a further set of non-imaged soil flow columns. For the imaged flow columns, the soil-water solution was extracted basally post-imaging and then UV–Vis analysis was immediately undertaken. The non-imaged flow column experiments were identical in set-up to the imaged columns (§2.6 and figure 1). However, prior to undertaking UV–Vis analysis on basally extracted soil-water solutions, the solutions were left to stand for 3 h in order for soil particles to precipitate out of suspension. The material which had not precipitated, including the AuNPs in suspension, was then extracted for centrifugation at 4000 RCF for 15 min using a Sorvall Legend XTR (Sorvall, UK). The supernatant was removed, the precipitated AuNPs were resuspended using UHQ to the initial concentration added to the columns (20 µl) and then UV–Vis analysis was undertaken. This centrifugation and resuspension would enable the concentration of AuNPs added at the start of the experiment to be compared with that of AuNPs extracted from the base of the columns at the end of the experiment.

## 2.5. Polymer coating process

The polyethylene glycol polymer used for the coating process was an α-methoxy-ω-mercapto PEG (CH3O-PEG-SH) with a molecular weight of 5 kDa (Rapp Polymere, France). A standard PEG coating protocol [28] was applied. To attach the PEG chains to the surface of the AuNPs, the PEG polymers were first dissolved in UHQ water at a concentration of 5 mg PEG/ml UHQ. A suspension of 100 nm AuNPs at a concentration of $3.84 \times 10^{-2}$ mg Au ml$^{-1}$ (Sigma Aldrich, UK) was then mixed with the PEG-UHQ solution at a ratio of 1 : 1. The mixture was wrapped in foil to avoid light exposure and then left to incubate for 1 h at room temperature. This suspension of PEG-AuNP complexes was then purified using a centrifugation method. For the purification, 80 ml of the suspension was split evenly across two 50 ml Falcon tubes with 40 ml in each tube. These were then centrifuged using a Thermo Scientific Heraeus Biofuge Primo R Centrifuge for 15 min at a force of 5000 RCF. Following centrifugation, the PEG-AuNP formed a pellet in the base of the centrifuge tube above which was a supernatant largely consisting of excess dissolved PEG and citrate buffer from the initial AuNP suspension. This supernatant was removed and the remaining pellet of PEG-AuNP was re-suspended using UHQ water to a volume of 20 ml per falcon tube. These falcon tubes were again centrifuged for 15 min at a force of 5000 RCF and the supernatant was removed though this time the pellet of PEG-AuNPs were resuspended to a total volume of 1 ml per Falcon tube. The two 1 ml volumes were then transferred into 1.5 ml Eppendorf tubes. Centrifugation was undertaken once more using the 1.5 ml Eppendorf tubes for 15 min at 4000 RCF. Following centrifugation 800 µl was removed from each of

the 1 ml mixtures to leave 200 µl of PEG-AuNPs at a concentration of 3.84 mg Au ml$^{-1}$—the concentration to be used in XCT imaging. Given that there were two Eppendorf tubes this led to a total output of 400 µl of PEG-AuNP suspension.

## 2.6. Assessment of gold nanoparticle stability in soil

To ascertain the influence of soil particle surfaces on PEG-AuNP transport and stability, a dynamic column transport experiment was set up. In this experiment, both single-material soil-analogues and the field soils were studied. Alumina (Al$_2$O$_3$) and silica (SiO$_2$) powders were chosen as initial substrates to examine the influence of 'worst case scenario' contrasting surface charges in the absence of salinity and pH effects. Silica has a point of zero charge at pH approximately 3 [29] while alumina has a point of zero charge at pH approximately 9 [30]. The PEG-AuNPs were anticipated to aggregate within this material, thus the intention was to characterize aggregation within a porous medium as a point of reference and comparison for PEG-AuNP stability in the subsequent soil column experiments. Any spatial pulse movement in the experimental system was observed and quantified using XCT imaging every 24 h and SEM imaging was used to characterize the interaction between silica, alumina and PEG-AuNPs in greater detail. The system was then also represented using a mathematical model to quantify a lower bound estimation for the buffer power of PEG-AuNPs in silica and alumina (this information is presented in electronic supplementary material, section 1).

Experimentally this system was constructed using 1 ml syringes for the column (figure 1). These syringes were held tip down in a vertical orientation. To prevent granular media from leaving the system, cotton wool was placed at the base of the syringe and lightly packed to fill a depth of 4 mm from the base. A silicon-rubber O-ring was then placed above the wadding to secure it in place (figure 1). For the soil-analogue experiment, the syringes were filled with either silica particles of 150–250 µm (Sigma Aldrich, UK), alumina particles of 50–200 µm (Fischer Scientific, UK), or an equal mixture of both to a depth of 4 cm above the O-ring. The SiO$_2$ and/or Al$_2$O$_3$ particles were then saturated by injecting UHQ water from the column base using a syringe pump (Harvard PHD 3000, Harvard, US) at a rate of 0.05 ml min$^{-1}$ until the meniscus was level with the upper surface of the granular media.

Once the column was fully saturated, a 20 µl pulse of PEG-AuNPs at a concentration of 3.84 mg Au ml$^{-1}$ was initialized at the top of the saturated material via pipette. This concentration, 100 times the concentration of the suspension as supplied from the manufacturer, was used to provide contrast in XCT assessments [10,11]. Another O-ring containing cotton wool in the central aperture was introduced directly above the PEG-AuNPs. This additional O-ring and cotton wool was to prevent flush-back of the PEG-AuNPs during the addition of 0.5 ml of UHQ water above the O-ring (figure 1). The UHQ water added above the O-ring provided the reservoir of fluid required for a constant-flux operation. The constant-flux was applied by drawing fluid through each column with the syringe pump at the pre-determined rate of 0.5 µm s$^{-1}$—the average maximum velocity of water movement through soil to plant roots [31]. The use of a syringe pump allowed the flux to be set to 0 µm s$^{-1}$ during imaging, reducing the incidence of movement artefacts, or 'blurring' and streaks, during image acquisition. A total of nine separate replicates of the system were manufactured, providing three replicates each of silica, alumina and the silica/alumina mixture. For the soil experiments, the syringes were filled with field soils in place of the soil-analogue powders. The experimental system with soil was otherwise identical. This soil column system was replicated nine times: three times for each of the three soils.

XCT imaging of the columns was undertaken once daily for a period of 3 days. After setting the flux to 0 µm s$^{-1}$, each sample was loaded into an XTH 225 L Industrial CT Scanner (Nikon, UK), and a 2D radiograph of the entire column was acquired at 40 kVp. A plot of average attenuation coefficient with distance down the column was produced to find the depth of the AuNP concentration peak maxima. A tomograph was acquired using a field of view centred on the peak including 2.4 mm of material either side of the peak position. The parameters for the XCT imaging are given in table 2. The reconstruction was undertaken using CTPro 3D (Nikon, Tokyo, Japan) using a fine scale dual centre of rotation detection.

Image processing of 3D reconstructions was carried out using FIJI [32]. First, a grey-level threshold was defined using the minimum and maximum grey values induced by nanoparticle accumulation (180–500 in a 32 bit image, as assessed across multiple positions in different scans). A 3D median filter was then applied using a kernel of 5 pixels in the $X$, $Y$ and $Z$ axes. This removed the influence of individual pixels of anomalously high grey value. The maximum grey value in each image slice down the column was then recorded and plotted. This was used to capture the nanoparticle pulse position.

**Table 2.** The parameters used for the X-ray CT imaging of the column based experimental system.

| parameter | value |
| --- | --- |
| voltage (kVp) | 40 |
| power (W) | 15 |
| exposure (ms) | 708 |
| analogue gain | 24 |
| binning | ×2 |
| number of projections | 1001 |
| frames per projection | 2 |
| target material | Mo |
| detector | Perkin Elmer 1621 Flat Panel |
| detector dimensions (pixels$^2$) | 2000 |
| pixel size on detector (µm$^2$) | 200 |
| detector dimensions (cm$^2$) | 40 |
| source to sample distance (mm) | 14.64 |
| source to detector distance (mm) | 755.03 |
| filtration | none |
| resulting pixel resolution (µm$^2$) | 8 |
| resulting scan time (min) | 25 |

Scanning electron microscopy (SEM) was used to further characterize the nature of AuNP and silica/alumina interaction following the column experiments. Samples were prepared by sectioning a 3 mm horizontal region of each syringe column which contained the AuNPs and the silica or alumina. These samples were air dried and then carbon coated using an Edwards Auto 306 Thermal Evaporator (Edwards, UK). The coating was between 15–20 nm thick of carbon and was applied using 'fine' pumping. SEM imaging was then undertaken in a full vacuum using a Leo 1450 VP SEM (Zeiss, UK) at a range of magnifications from 50 to 30 000×.

## 2.7. Confirmation of AuNP XCT contrast and stability

This experimental system consisted of 1 ml syringes (BD Plastipak, UK) that were cut halfway down (at the 0.5 ml graduation mark). The basal halves (syringe tip end) were held tip down with a rubber bung placed inside at the base and the top region of each syringe was discarded. Above this basal bung, these syringes were filled to the top with soil which was fully saturated with a suspension of PEG-AuNPs at the concentration of 3.84 mg Au ml$^{-1}$. The syringes were then transferred into the XTH 225 L Industrial CT Scanner and underwent XCT imaging using the same imaging parameters as described in table 2.

## 2.8. Assessing the utility of gold nanoparticles as a tracer

### 2.8.1. Experimental imaged system

The experimental set-up for estimating nanoparticle diffusivity consisted of a 1 ml syringe held tip down and connected to a syringe pump at the basal tip (Harvard PHD 3000, Harvard, US). Within the syringe Bangor soil sieved to between 1.18 and 2 mm was placed until it had achieved a depth of 3 cm (figure 2). The Bangor soil was then fully saturated with AuNPs at a concentration of 2.5 mg Au ml$^{-1}$. This concentration was distinguishable from UHQ water and soil but without the risk of producing considerable artefacts within the XCT images that might have been produced at higher AuNP concentrations (for example those present in figure 7). Above the soil fully saturated with AuNPs, UHQ was added to the top of the system until a depth of 3 cm was achieved.

This system was XCT imaged every hour for 3 h. Between images the syringe pump applied a flux of 1 µl min$^{-1}$ for a duration of 30 min. The XCT parameters used for this imaging were the same as those for

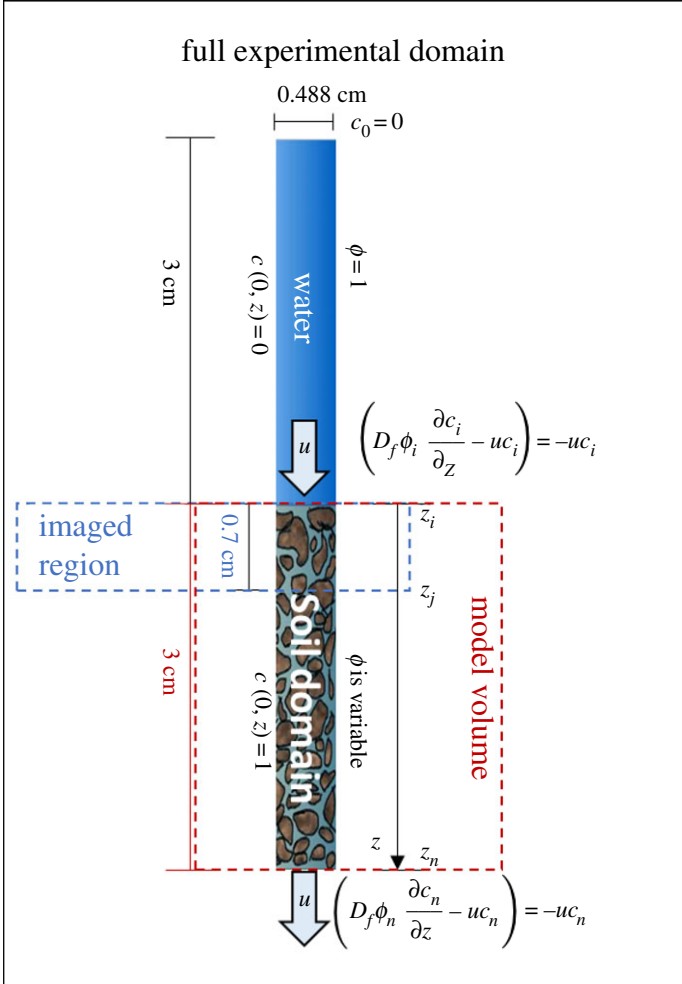

**Figure 2.** Modelling experiment of gold nanoparticles as a flow tracer. From the entire experimental domain, the simulations only consider the soil subdomain. The model uses a zero gradient at the top and bottom boundaries of the soil domain considering the full experiment. The model is used to fit data obtained from the imaged region (the top 0.7 cm of the soil domain) in order to estimate the diffusivity of gold nanoparticles in free water.

previous experiments (table 2). The region of the experimental system that was imaged during XCT imaging was from the surface of the soil to 0.7 cm beneath the soil surface. This would enable the 'transition' zone, the region in which the advection and diffusion at the soil-water interface could be observed, to be captured in detail.

These images were kept as 32 bit images and not downscaled to 8 bit such that greater grey value range could be maintained. The image processing for this dataset consisted of cropping the image such that only the inside of the syringe remained. A grey value threshold of between 220 and 360 was applied leaving only the liquid phase (AuNP suspension/ water) within the image. The area and mean average grey value of each vertical z slice was then recorded to provide the porosity and average grey value, respectively, of the liquid phase with depth.

### 2.8.2. Modelling of gold nanoparticle advection–diffusion

In order to assess the utility of using the gold nanoparticles as a flow tracer, the nanoparticles diffusivity required quantification. The experimental system was modelled with a focus on the soil domain (figure 2). The soil domain is considered to be fully water saturated. Nanoparticle concentrations in the soil pore space are used based on the normalized average image grey value intensities for a slice at a given depth value, assuming the grey values to be proportionate to nanoparticle concentrations.

The concentrations are normalized based on the following formulation:

$$c(t,z) = \frac{\tilde{c}(t,z) - \text{mean}(\tilde{c}(t_f))}{\text{mean}(\tilde{c}(t_0)) - \text{mean}(\tilde{c}(t_f))}, \tag{2.1}$$

where $\tilde{c}(t,z)$ (mol m$^{-3}$) is the solute concentration at a given time and depth, $t_0$ (s) is the initial time of the experiment, $t_f$ (=7200 s) is the final time of the experiment, and mean $(\tilde{c}(t_0))$ and mean $(\tilde{c}(t_f))$ (mol m$^{-3}$) are the mean values of the concentration across the domain at the initial and final time points respectively. The mean values are calculated as:

$$\text{mean}(\tilde{c}(t_k)) = \frac{1}{n-i} \sum_{j=i+1}^{n} \tilde{c}(t_k, z_j), \tag{2.2}$$

where $z_i$ (m) is assumed to be the soil surface and $z_n$ (m) is the bottom boundary of the soil domain (figure 2). Transport of gold nanoparticles in the water phase was modelled based on the advection–diffusion equation:

$$\frac{\partial(\phi c)}{\partial t} + \frac{\partial}{\partial z}(uc) = \frac{\partial}{\partial z}\left(D_f \phi \frac{\partial c}{\partial z}\right), \quad z \in [z_i, z_n], \tag{2.3}$$

where $\phi[\text{m}^3_{\text{fluid}} \text{m}^{-3}_{\text{bulk}}]$ is the soil porosity, $c$ (–) is the normalized concentration of gold nanoparticles, $u[\text{m}^3_{\text{fluid}} \text{m}^{-2}_{\text{surface}} \text{s}^{-1}]$ is the flux value governed by the pumps flow rate and the syringe cross-sectional area, and $D_f$ (m$^2$ s$^{-1}$) is the diffusivity of gold nanoparticles in free water. The water flux is also assumed constant throughout the soil domain. The porosity is considered time independent. In the experiment, water is pulled through the soil domain, ultimately flushing the gold nanoparticles out of the soil system. Considering all of the experimental constraints, the model, initial conditions and boundary conditions could be expressed as:

$$\begin{cases} \phi \dfrac{\partial c}{\partial t} = \dfrac{\partial}{\partial z}\left(D_f \phi \dfrac{\partial c}{\partial z} - uc\right), & z \in [z_i, z_n] \\[2mm] c = 1, & t = 0 \\[2mm] \dfrac{\partial c}{\partial z} = 0, & z = z_i \\[2mm] \dfrac{\partial c}{\partial z} = 0, & z = z_n \end{cases} \tag{2.4}$$

Two porosity scenarios are considered when running the simulations. The first scenario considers a fixed average porosity throughout the soil domain of $\phi = 0.4\,\text{m}^3_{\text{fluid}}\,\text{m}^{-3}_{\text{bulk}}$. The second scenario considers a spatially varying porosity for the first 0.7 cm depth based on the experimentally determined porosity values at each slice. For both scenarios, the model is fit to the data based on an optimization routine by treating the diffusivity as a fitting parameter. The objective function is created based on the sum of the squared difference between the model results and the data points:

$$\epsilon = \sum_{k=0}^{f} \sum_{j=i}^{n} (c_{\text{mod}}(t_k, z_j) - c_{\text{dat}}(t_k, z_j))^2, \tag{2.5}$$

where $c_{\text{mod}}$ (–) are the modelled normalized concentration values, $c_{\text{dat}}$ are the normalized experimental concentrations, $t_k$ (s) is a given time point and $z_j$ (m) is the depth of a given slice. Numerical solution and optimization were all conducted using Matlab 2016a (Mathworks, Inc., Natick, MA, USA [33]) via a custom built Crank–Nicolson scheme for solving the PDEs and using the fmincon function for the optimization.

# 3. Results

## 3.1. Characterizing soil-water solutions

The pH values for the Bangor, Dundee and Nottingham soil-solutions stored at room temperature were 7.12, 7.25 and 7.83, respectively. For the refrigerated soil-water solutions, the pH values were 6.33, 6.72 and 7.19, respectively. Therefore, all soil-water solutions were of near-neutral pH and there was only minimal variation in pH between the three soils, but Dundee and Bangor soils were slightly more

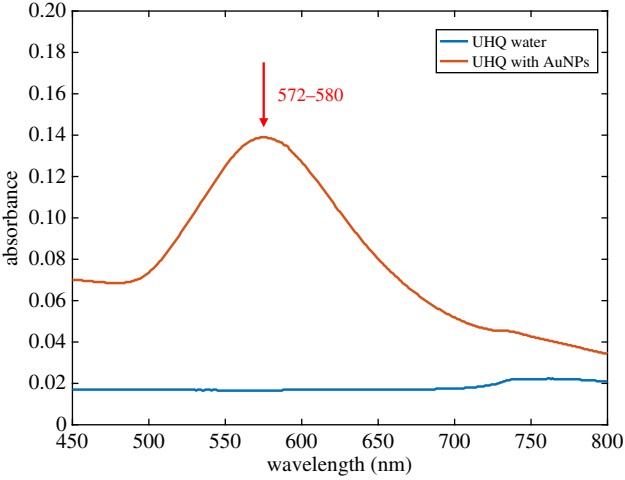

**Figure 3.** The characteristic absorbance peak of 100 nm AuNPs in UHQ water using UV–Vis analysis. The blue line represents UHQ water without the addition of AuNPs and the orange line represents UHQ water 24 h after the addition of AuNPs. The red arrow and adjacent numeric values indicate the AuNP peak position.

**Table 3.** The selected results of ICP-MS assessment of soil-water solutions. Elements displayed are those which display variation in concentration between soils.

| soil | element | | | | | | |
| --- | --- | --- | --- | --- | --- | --- | --- |
| | Mg (ppb) | Al (ppb) | K (ppb) | Ca (ppm) | Fe (ppb) | Rb (ppb) | Ce (ppb) |
| Bangor | 2579 | 28 | 1658 | 13 | 25 | 0.56 | 0.14 |
| Dundee | 307 | 328 | 2969 | 1 | 394 | 1.34 | 0.43 |
| Nottingham | 3571 | 36 | 2126 | 16 | 35 | 0.72 | 0.17 |

acidic than the Nottingham soil. The ICP-MS results suggested that the Dundee soil contained higher concentrations of aluminium, iron, magnesium, potassium, rubidium and caesium (table 3). Additionally, the Bangor and Nottingham soils contained higher levels of calcium. These elemental concentrations may influence the behaviours of AuNPs in these three different soils.

## 3.2. Gold nanoparticle stability in soil-water solutions

UV–Vis analysis of AuNPs with UHQ water indicated the characteristic peak for stable 100 nm AuNPs between wavelengths 572 and 580 nm (figure 3). This spectrum was used as the reference for nanoparticle stability against which the UV–Vis spectra of AuNPs suspended in soil-water solutions were compared. These samples also allowed determination of erroneous background peaks.

When NC-AuNPs were added to soil-water solutions the UV–Vis spectroscopy results revealed that the NC-AuNPs became destabilized in all instances (figure 4a,c,e). The NC-AuNPs became destabilized within the Bangor and Nottingham soil-water solutions before 24 h and within the Dundee soil-water solution between 24 and 72 h. This destabilization is indicated by the disappearance of the peak between the wavelengths 572 and 580 nm, when compared to the AuNP reference spectrum (figure 3). The increase in absorption between wavelengths of approximately 200 to 500 nm, present in all soil solutions, is likely noise resulting from dissolved organic carbon in the soil-water solution [34,35].

Since the NC-AuNPs are becoming destabilized by the soil-water solutions, the AuNPs underwent the coating procedure with a 5 kDa PEG-SH polymer. The UV–Vis experiment was then repeated using the PEG-AuNPs. In this instance the peak between wavelengths of 572 and 580 nm persisted, and the relative concentration of 100 nm AuNPs (as indicated by background and peak height ratio) remained constant over a 7 day period (figure 4b,d,f). This suggested that the PEG-AuNPs had remained stable.

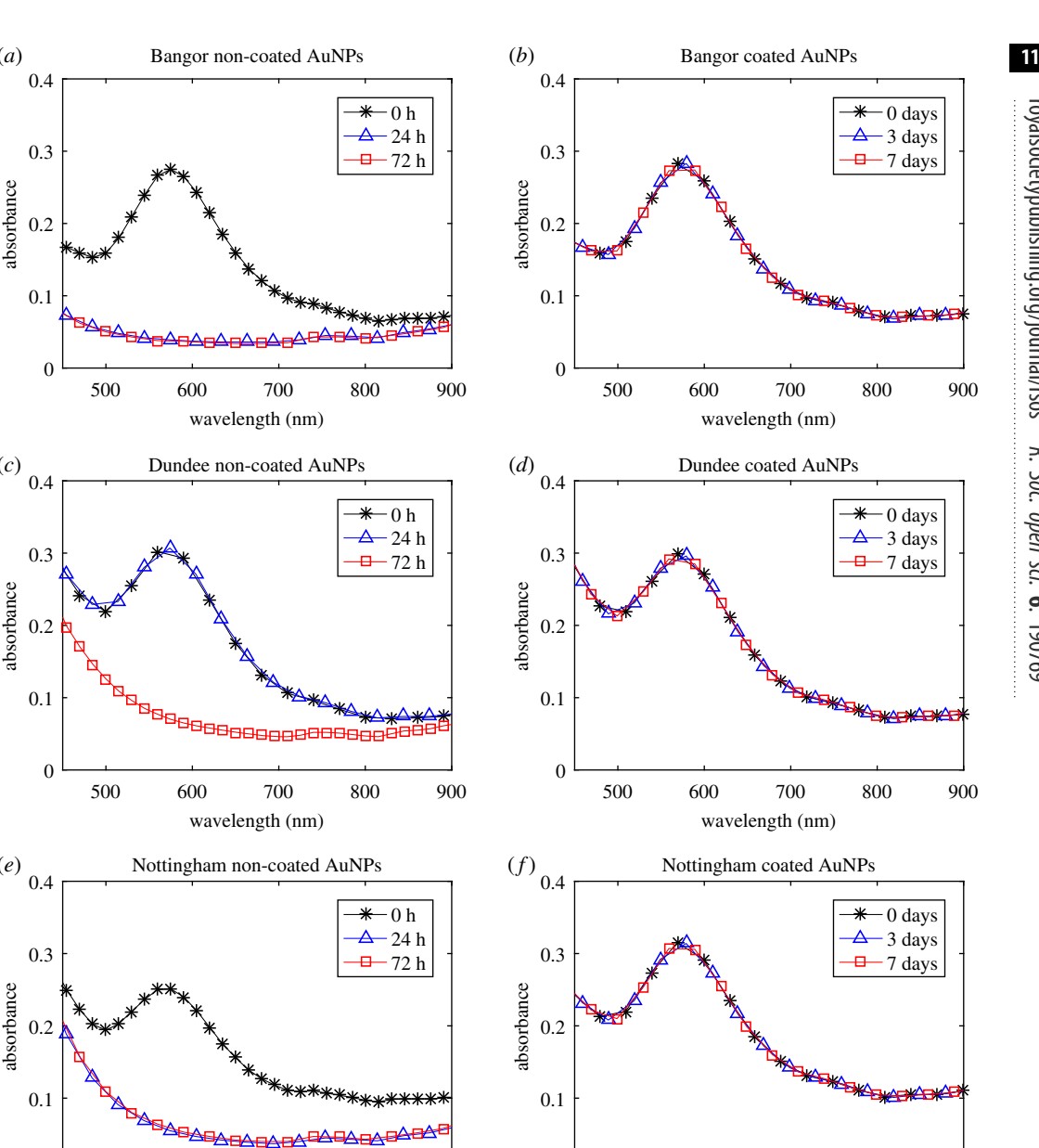

**Figure 4.** Confirmation of the stability of PEG coated AuNPs in soil-water solutions of the Bangor, Dundee and Nottingham soils using UV–Vis analysis. The UV–Vis analyses of non-coated AuNPs in Bangor (*a*), Dundee (*c*) and Nottingham (*e*) soil solutions at time steps of zero, 24 and 72 h from initial AuNP addition, are on the left side. The UV–Vis analyses of PEG coated AuNPs in Bangor (*b*), Dundee (*d*) and Nottingham (*f*) soil solutions at time steps of 0, 3 and 7 days from initial AuNP addition, are on the right side.

## 3.3. Assessment of gold nanoparticle stability in porous media

### 3.3.1. Silica and alumina experimental results

XCT images of silica and alumina columns demonstrated that destabilization of PEG-AuNPs was primarily characterized by almost immediate accumulation of the gold nanoparticles on both silica and alumina surfaces in all experiments. This accumulation occurred within an hour in all cases and was detectable only 1 h after the addition of the PEG-AuNPs to the column. The accumulation was observed in the top few millimetres of each column as a layer of high grey value voxels on the exterior surfaces of particles (electronic supplementary material, figures S1 and S2). These voxels were of such high grey value that they can be segmented from silica and alumina using a simple grey value threshold (figure 8). In fluid-filled pore spaces immediately adjacent to the silica and alumina

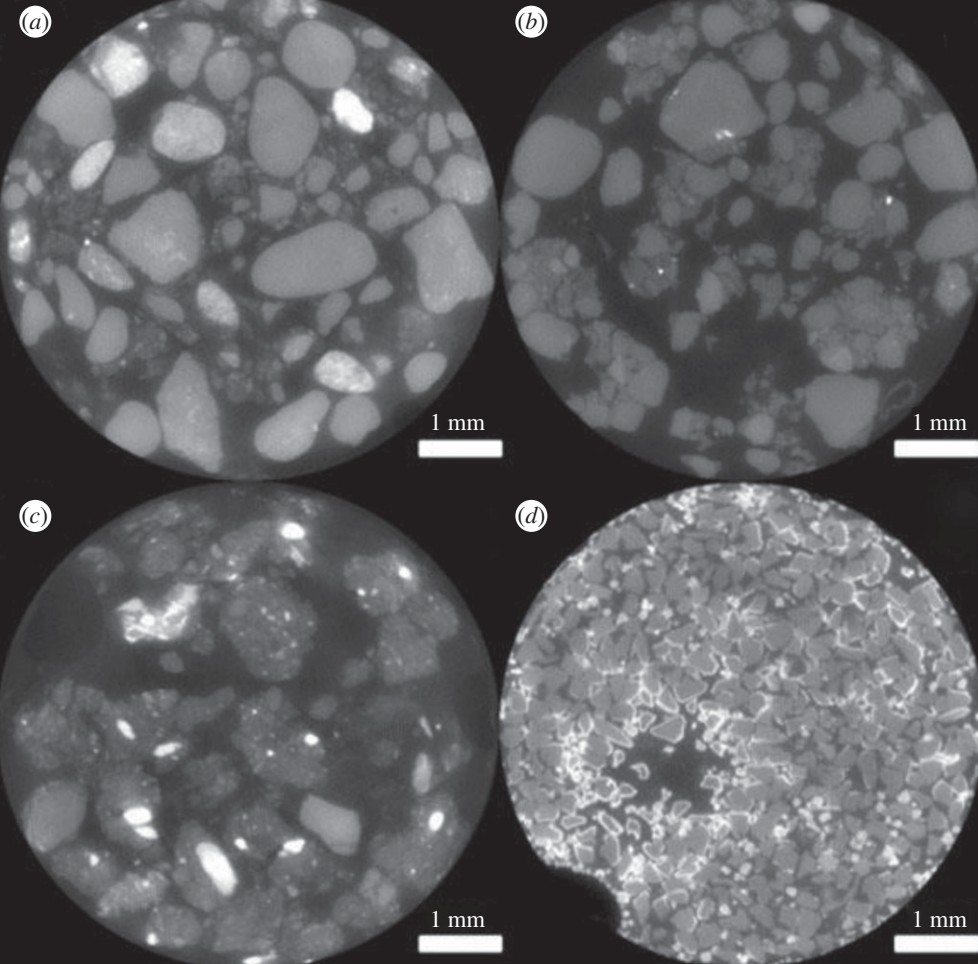

**Figure 5.** Example image slices from XCT images of the PEG coated AuNP flow columns of Bangor (*a*), Nottingham (*b*), Dundee (*c*) and silica/alumina mixture (*d*). The silica/alumina mixture is included as a point of reference for the expected appearance of destabilization. The surface deposition visible in the silica and alumina mixture is not present in the soil images suggesting aggregation has not taken place.

particles that had AuNPs accumulated on their surface, grey levels were similar to pure UHQ water. This indicates a significant drop of suspended concentrations of AuNPs within the solution after this 1 h period.

The SEM images of the silica and alumina displayed the accumulation of the AuNPs on the silica and alumina particles in greater detail. The AuNPs appear to be unevenly accumulated and generally on the uppermost exposed surfaces (electronic supplementary material, figure S3). At the highest magnification (30 000×) it was observed that both individual and aggregated groups of nanoparticles were present on the alumina and silica surfaces (electronic supplementary material, figure S3d).

### 3.3.2. Soil experimental results

The columns containing each of the three soils were imaged every day for the full 3 days and displayed no indication of the AuNP aggregation observed within the silica and alumina columns (figure 5). Visibly the PEG-AuNPs could be seen passing down through the column between imaging time steps, pulled by the syringe pump. XCT images showed that there were no brighter regions encapsulating primary soil grains, though there is the appearance of more X-ray attenuating materials inside grains—particularly in the Dundee soil (figure 5*c*). However, these brighter regions are most likely naturally occurring minerals within the soil grains.

The results of the UV–Vis undertaken on the solutions extracted from the soil columns displayed the characteristic 100 nm AuNP absorption peak between the wavelengths of 572 and 580 (figure 6). This peak is still present within the XCT imaged Nottingham soil sample (figure 6*a*), however it is visible

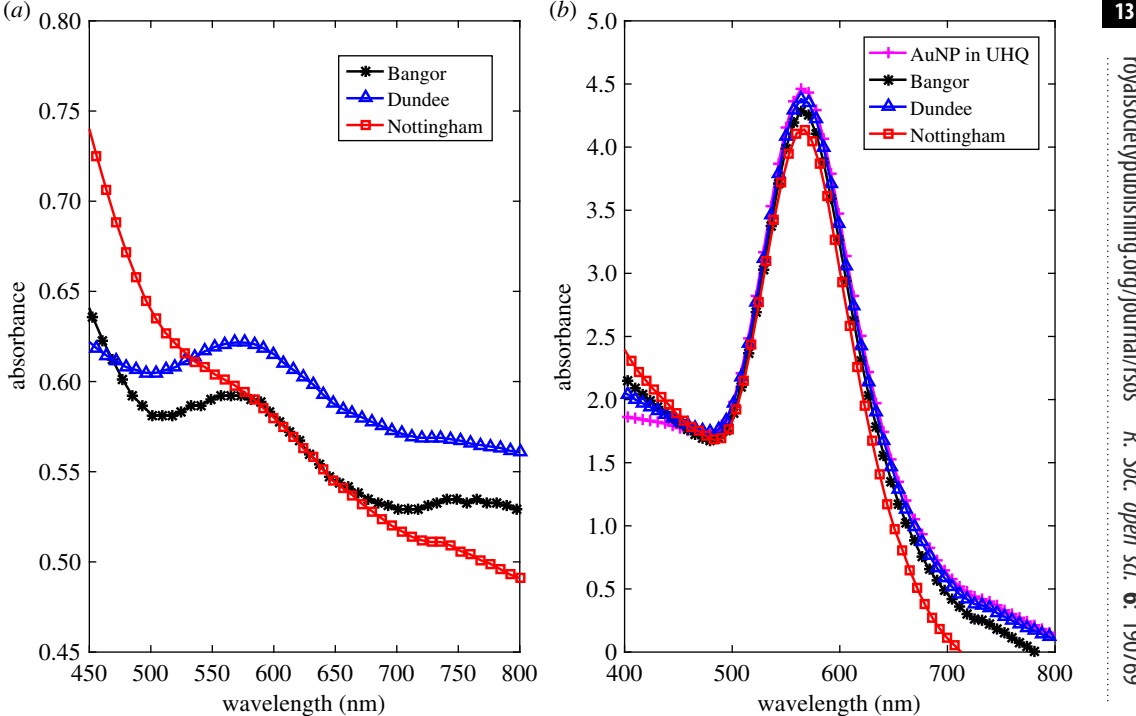

**Figure 6.** The presence of stable PEG coated AUNPs in soil-water solutions extracted from soil columns as demonstrated by the results of UV–Vis analysis. This analysis was undertaken on solutions extracted from columns of each of the three soils (Bangor, Dundee and Nottingham). This included those columns which were XCT imaged (*a*) and those flow columns which were run exclusively for UV–Vis analyses (*b*). The peaks between wavelengths 572 and 580 nm indicates the presence of 100 nm gold nanoparticles which had remained stable in suspension. The appearance of peaks at wavelengths below 500 nm is most likely the result of small soil particles within the suspension that were not removed by filtration.

as a 'shoulder' within the spectrum. These UV–Vis results contained more background noise than previous UV–Vis data—likely caused by small soil particulates which were not removed by filtration. Unlike previous soil-water solutions, these solutions could not undergo such thorough filtration due to the risk of also removing the AuNPs.

## 3.4 Confirming AuNP XCT contrast and stability

In the XCT images of the soils saturated with PEG-AuNPs, the contrast between the soil and PEG-AuNP solution was evident (figure 7). The PEG-AuNP suspension was visibly more attenuating than the soil phase and grey level histograms of regions containing both soil and PEG-AuNP phases display a bi-modal distribution. The peak value in the histograms for the soil phase in the 8 bit images was at a pixel value of approximately 89 and for the PEG-AuNP suspension the peak is at 124. The contrast between PEG-AuNPs and soil means that the PEG-AuNP suspension can also be segmented using only a simple grey value threshold (figure 8). There were artefacts present within the images, which are most likely due to beam hardening resulting from the greater attenuation of PEG-AuNPs. There is also no evidence of the formation of bright rings around soil particles as was visible in the silica and alumina images.

## 3.5. Estimated gold nanoparticle diffusion rate

Experimentally the image results suggest we were able to capture the movement of AuNPs and ingress of water into the imaged domain. This is captured in the declining grey values with depth between time steps (figure 9) following image processing. The model was then fit to this image data.

Results for model fitting are illustrated in figure 9. For the scenario considering a constant porosity (figure 9*a*), the optimization suggests that the diffusivity of the gold nanoparticles in free water is $D_f = 6.5 \times 10^{-9}\,\mathrm{m^2\,s^{-1}}$. The diffusivity reduced to $D_f = 2.5 \times 10^{-9}\,\mathrm{m^2\,s^{-1}}$ when considering the actual porosity at each depth (figure 9*b*). Both diffusivity values are on the same order of magnitude of

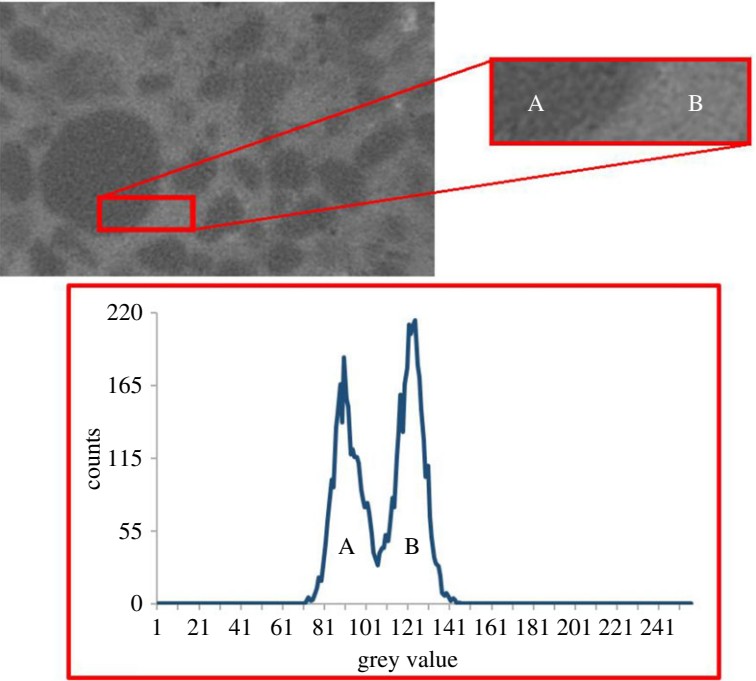

**Figure 7.** The XCT contrast provided by a suspension of PEG-AuNPs within a syringe barrel containing soil particles. Top left: an XCT image region of a PEG coated gold nanoparticle suspension, at a concentration of 3.84 mg Au ml$^{-1}$, fully saturating Bangor soil. Top right: the PEG coated gold nanoparticle suspension (B) is visibly more attenuating than the soil (A). However, there were artefacts present within the images, which are most likely due to beam hardening resulting from the greater attenuation of PEG-AuNPs. Bottom: this contrast between the attenuation of soil (A) and PEG coated gold nanoparticles (B) in the top right image is also visible in the histogram of grey values.

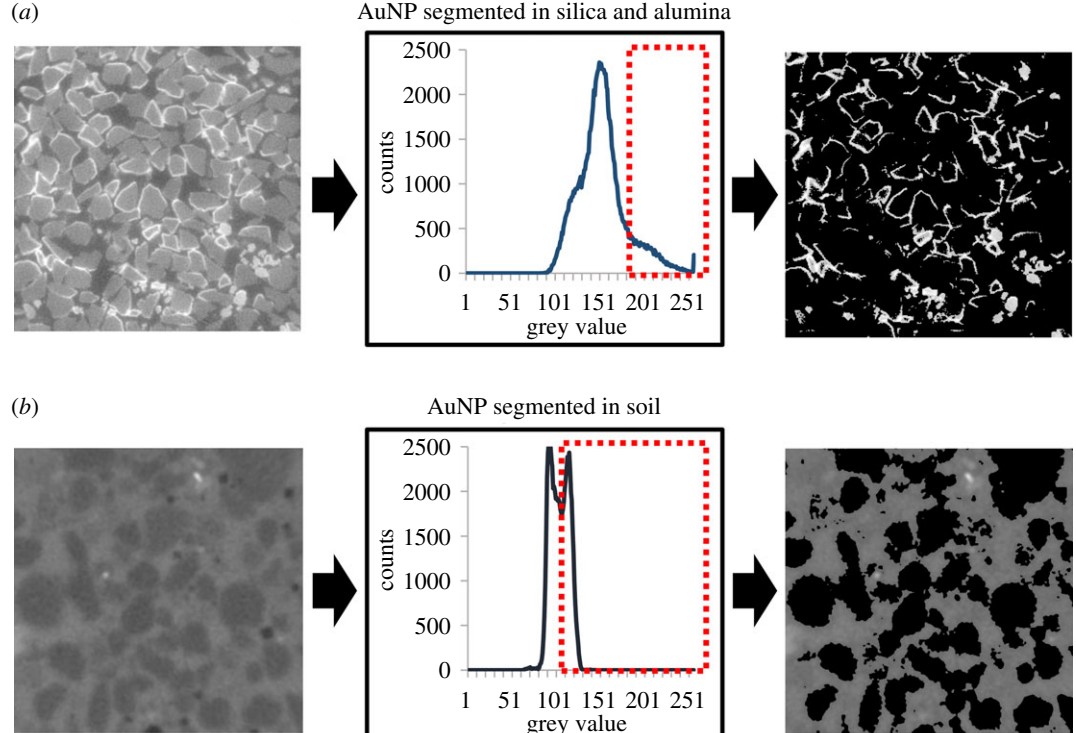

**Figure 8.** A demonstration that AuNPs can be segmented from silica and alumina (top row) and from soil (bottom row) using a simple grey value threshold. The images on the left are unaltered 8 bit images. In the centre are the histograms of grey values for the unaltered images. The red boxes with the dashed outline indicate the grey value range of AuNPs which would be segmented by a grey value threshold. The images on the right are the segmented AuNPs and the result of the grey value threshold.

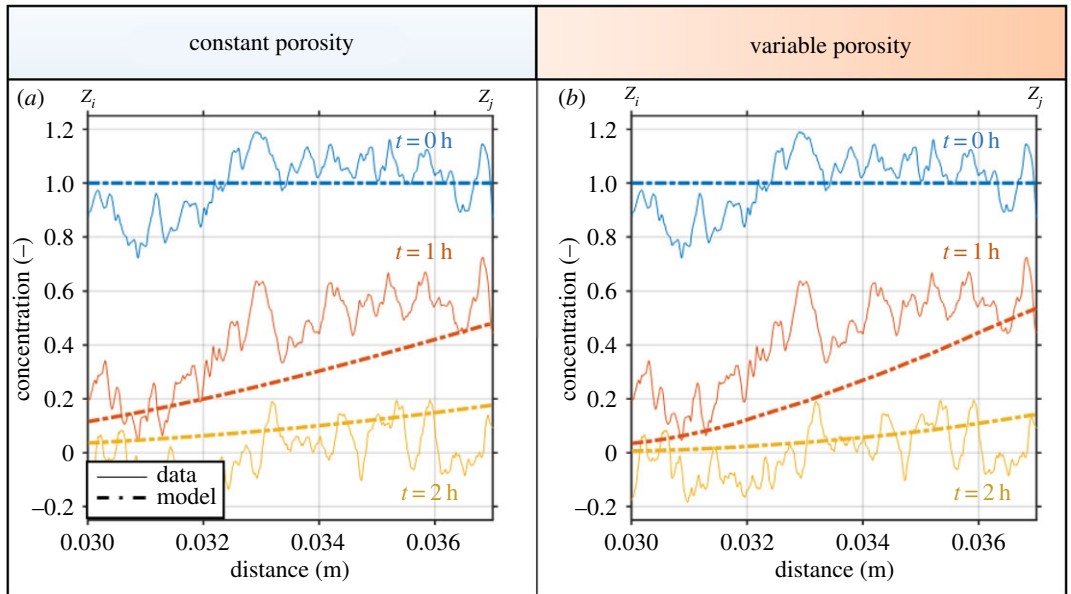

**Figure 9.** Modelled and experimental gold nanoparticle concentrations along the depth of the imaged region considering (a) a constant value for porosity and (b) a spatially varying porosity. Experimentally determined normalized concentrations are given as solid curves, while modelled normalized concentrations are given as dashed curves. The model was fit for three different time points (0, 1 and 2 h), which are colour coded as blue, orange and yellow respectively. Fit curves to the data estimate that nanoparticles have a diffusivity in free water of (a) $6.5 \times 10^{-9}$ m$^2$ s$^{-1}$ for a fixed porosity value of $\phi = 0.4$ m$^3_{\text{fluid}}$ m$^{-3}_{\text{bulk}}$, and (b) $2.5 \times 10^{-9}$ m$^2$ s$^{-1}$ for a spatially varying porosity.

other typical plant nutrients diffusing in free water [36]. Simulated results suggest that nanoparticles diffuse into the free water domain prior to being flushed out of the system. Furthermore, the nanoparticles do not appear to be completely flushed out of the system at the final time point.

# 4. Discussion

This study investigated the stability of AuNPs in soil environments. The conditions provided by soil-water solutions were confirmed by UV–Vis to induce aggregation of NC-AuNPs. This effect was mitigated by applying a PEG-SH coating to the AuNPs which proved sufficient for maintaining stability within soil-water solutions. When applied to flow columns of silica and alumina the destabilization of AuNPs in porous media was characterized using a combination of XCT and SEM imaging techniques. The PEG-AuNPs were then added to flow columns containing soils where it was confirmed using XCT and UV–Vis that the PEG-AuNPs had remained stable. This stability was further demonstrated in the columns where soil was fully saturated with PEG-AuNPs and which provided sufficient distinct contrast.

The ICP-MS analysis of soil solutions indicated that the Dundee soil likely had the highest clay content of the three soils. This is because the Dundee soil contained higher concentrations of metals associated with clay-minerals (table 3) such as aluminium, iron, magnesium and alkali metals such as potassium, rubidium and caesium [37]. Additionally, the ICP-MS results suggested the Bangor and Nottingham soils may contain more sand minerals than the Dundee soil, as evidenced by the higher calcium content [37]. It is possible that these variations in soil composition meant that the Dundee soil contained different soluble salts than the other two soils. Given the known susceptibility of NC-AuNPs to destabilizing conditions [12], it was expected that they would undergo aggregation in the soil solutions, observable as a peak shift in UV–Vis analysis (figure 4a,c,e). This could have meant the salts within the Dundee soil solution had a less severe destabilizing effect than those of the other two soils hence the NC-AuNPs remained stable for longer within this soil solution—up to 72 h (figure 4c). This reduced aggregation was unlikely to be acidity-related, since all three soil solutions were relatively neutral in pH and an absence of soil particles indicates that the differences were not related to surface charge effects.

In all cases the NC-AuNPs remained stable for less than 72 h. In contrast, the PEG-AuNPs introduced to soil solutions remained stable for at least one week (figure 4*b,d,f*). This indicated that the 5 kDa PEG-SH coating was sufficient to prevent aggregation in the presence of soluble soil salts and acids. The success of PEG-AuNPs for preventing aggregation is also corroborated by the findings of Sun *et al.* [10], who observed that PEG coated nanoparticles remained stable at concentrations of 100 mg Au ml$^{-1}$ and Bergen *et al.* [15] who noted that PEG coating prevented destabilization from mild saline and acidic conditions.

The XCT images of the soil-analogue columns demonstrated that both silica and alumina (with contrasting negative and positive charges) had compromised the stability of the PEG-AuNP suspension as expected. The experimental results indicated significant and rapid immobilization, suggesting that adhesion occurred within an hour. The XCT images used to characterize the destabilization of the PEG-AuNPs acted as a reference for comparison against the soil column images (electronic supplementary material, figures S1 and S2). Where AuNPs had become immobilized, they were visible as bright regions on the surface of silica and alumina particles in XCT images. These regions provided considerable contrast (electronic supplementary material, figures S1 and S2) and also meant the PEG-AuNPs could be segmented from the silica/ alumina using a grey value threshold (figure 8). The XCT images of soil-analogue columns and the significant contrast provided by accumulated AuNPs did however demonstrate the potential of AuNPs as contrast agents within porous media systems. If AuNPs could in future be developed to accumulate on target structures, for example root systems, they could be a useful tool in root/ soil XCT studies.

The SEM imaging enabled further characterization of nanoparticle destabilization. The resulting images indicated that the accumulation was not uniform with AuNPs binding on the uppermost surface. Furthermore, SEM imaging showed that the nanoparticles did not necessarily aggregate with one another (electronic supplementary material, figure S3). This further supports the observation that the immobilization was fast as the suspension was not able to penetrate through the column beyond the uppermost surface.

The XCT images of the columns of soil were compared against those of silica and alumina columns. The bright surface coatings that were characteristic of AuNP aggregation in the silica and alumina columns were notably absent within the soil column images. Additionally, whereas the AuNPs were visible by eye on the surface of the silica and alumina as a dark brown region at the surface, within the soil columns the PEG-AuNP suspension could be seen moving further down the column between imaging time steps. XCT images of the soils without the addition of AuNPs indicated that the bright contrasting regions embedded inside Dundee primary soil grains (figure 5*c*) are likely regions of naturally occurring soil minerals with a high X-ray attenuation rather than AuNPs. Not only were the primary soil grains not permeable enough for AuNP infiltration, but there is also no indication of AuNP accumulation on external soil grain surfaces, as had been observed in the silica and alumina. Given the complexity of the chemical environment provided by soil and the variability in soil mineral attenuation it would likely be worthwhile to undertake future investigations initially using sand. This would provide a less complex chemical environment and the X-ray attenuation of sand mineral grains would likely be more homogeneous.

The reason the AuNPs did not produce contrast while in suspension against soil or water, despite the initial high concentration, is likely because a small volume of water (approximately 10 µl) remained above fully saturated soil prior to the addition of the AuNPs. Once the AuNPs were applied to the surface they were likely diluted by this water volume as the high velocity with which the droplet of AuNPs entered the water may have been turbulent. Hence this would explain why the AuNP suspension became undetectable in XCT at a faster rate than standard diffusion would anticipate. This phenomena has been demonstrated by Taylor [38], Fischer [39] and Rein [40]. These imaging results therefore suggested that the PEG-SH coating had enabled AuNPs to remain stable in the soil columns.

UV–Vis results for solutions extracted from both the imaged and non-imaged soil columns confirmed that 100 nm PEG-AuNPs in suspension remained stable after the 3 days of imaging (figure 6) for all three soil types. This was indicated by the presence of the characteristic absorption peak between the wavelengths of 572 and 580. In particular, the absorption values for the characteristic AuNP peaks in figure 6*b* demonstrate this AuNP stability. This is because the concentration of AuNPs extracted from the base of the columns was almost equal to that of the AuNP suspensions applied to the soil surface in the column 3 days earlier. The increased noise present within this UV–Vis data was likely the result of small soil particles which could not be removed from the solution by filtration or dissolved organic carbon. The region below a wavelength of 500 nm partially obscured the characteristic peak between 572 and 580 nm in the Nottingham spectra in the imaged columns as a 'shoulder' (figure 6*a*) and this is an example of such noise.

Soil fully saturated with PEG-AuNPs provided further confirmation that the PEG coating ensured stability (figure 7). The bright rings of destabilized AuNPs that were visible on the surface of silica and alumina were not present in these samples. In addition, the X-ray attenuation of the PEG-AuNP suspension appeared uniform and was greater than that of the soil particles. The contrast between the PEG-AuNPs and the soil particles also meant that the PEG-AuNPs can be easily segmented using only a simple grey value threshold (figure 8). This uniformity indicates that the PEG-AuNPs remained stable in suspension in the presence of the soil particles even at high concentration. There did however appear to be artefacts within the image most likely resulting from beam hardening caused by the greater attenuation of PEG-AuNPs. The contrast provided by the PEG-AuNP suspension and the bi-modal nature of the contrast between the PEG-AuNPs and soil particles also demonstrated that the 3.84 mg Au ml$^{-1}$ concentration of nanoparticles was sufficient for contrast against soil grains when not exposed to dilution. Furthermore, this result reinforces the suitability of AuNPs for future development as contrast agents in soil systems.

The diffusivity of the gold nanoparticles in free water was estimated as $D_f = 2.5 \times 10^{-9}$ m$^2$ s$^{-1}$ when the actual porosity was considered at each depth (figure 9b) indicating that PEG-AuNPs diffuse at a similar rate to other typical plant nutrients [36]. This indicates that they may diffuse too quickly to be effective as a tracer contrast agent given the high concentration that is required for producing sufficient contrast. However, for use as a proposed functionalized contrast agent in soil PEG-AuNPs remain a promising candidate.

In summary, the PEG coating applied to the AuNPs was sufficient to maintain stability of the nanoparticles in soils. This is evidenced by both UV–Vis and XCT imaging results. The stabilization of AuNPs for use in soil environments can act as a platform from which development of AuNPs as XCT contrast agents can be pursued. While there are limitations to the use of AuNPs as 'bulk flow' contrast agents (such as the high required concentration and estimated diffusion rate in free water) there is evidently potential for use in applications where AuNPs accumulate in high concentrations at a target surface, which may be enabled through functionalization.

# List of abbreviations

AuNP(s)      gold nanoparticle(s)
NC-AuNP(s)   non-coated gold nanoparticle(s)
PEG-AuNP(s)  PEG coated gold nanoparticle(s)
ICP-MS       inductively coupled plasma mass spectrometry
UHQ          ultrapure water (water purity Type 1 as described by ISO 3696)
PEG          polyethylene glycol
SEM          scanning electron microscopy
UV–Vis       ultraviolet–visible spectroscopy
XCT          X-ray computed tomography

Data accessibility. Data supporting this study is publicly available from the University of Southampton repository via the doi: https://doi.org/10.5258/SOTON/D0882.

Authors' contributions. C.P.S., A.v.V., S.D.K., I.E.D. and T.R. designed the study. C.P.S. collected the UV–Vis data. C.P.S. and A.v.V. collected the ICP-MS data. C.P.S. and S.D.K. collected the XCT data. M.M.-H. and I.E.D. developed the PEG coating method. M.M.-H., I.E.D. and C.P.S. undertook the coating procedure. C.P.S. collected the SEM data. S.J.D. and S.A.R. developed the modelling. C.P.S. wrote the manuscript and all other authors provided critical revision and approval before submission and publication.

Competing interests. We have no competing interests.

Funding. C.P.S., A.v.V., S.D.K. and T.R. are funded by ERC Consolidator grant no. 646809 (Data Intensive Modelling of Rhizosphere Processes). S.A.R. and T.R. are funded by BBSRC SARISA BB/L025620/1. T.R. is also funded by BBSRC SARIC BB/P004180/1, EPSRC EP/M020355/1 and NERC NE/L00237/1. S.J.D. is funded by BBSRC Syngenta Case PhD Studentship BB/L5502625/1.

Acknowledgements. C.P.S., A.v.V., S.D.K. and T.R. are funded by ERC Consolidator grant no. 646809 (Data Intensive Modelling of Rhizosphere Processes). S.A.R. and T.R. are funded by BBSRC SARISA BB/L025620/1. T.R. is also funded by BBSRC SARIC BB/P004180/1, EPSRC EP/M020355/1 and NERC NE/L00237/1. S.J.D. is funded by BBSRC Syngenta Case PhD Studentship BB/L5502625/1. The authors acknowledge the µ-VIS X-ray Imaging Centre at the University of Southampton for provision of tomographic imaging facilities, supported by EPSRC grant no. EP-H01506X. The authors acknowledge the Scanning Electron Microscope Facility and High Resolution

ICP-MS Facility at the National Oceanography Centre Southampton. The authors would also like to acknowledge Davey Jones (Bangor University), Tim George and Glyn Bengough (James Hutton Institute) and Sacha Mooney and Malcolm Bennett (University of Nottingham) for providing the soil samples used within this investigation.

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
