## [Reviewer comments · Royal Society Open Science]

Review History

RSOS-190769.R0 (Original submission)

Review form: Reviewer 1

Is the manuscript scientifically sound in its present form?

Yes

Are the interpretations and conclusions justified by the results?

Yes

Is the language acceptable?

Yes

Do you have any ethical concerns with this paper?

No

Have you any concerns about statistical analyses in this paper?

No

Recommendation?

Accept with minor revision (please list in comments)

Comments to the Author(s)

The manuscript provides evidence of stabilization of gold nanoparticles (preventing their aggregation) for application as x-ray contrast agent in soils. After showing that the nanoparticles aggregate when not coated preventing their applicability for day-long experiments, the team coated the nanoparticles with PEG, and achieved stability for a period of several days. To further confirm their applicability for soils, they used time-resolved x-ray CT combined with image-based modeling to evaluate nanoparticle diffusion in wet soil.

I think the paper would stand as an important contribution to the x-ray imaging and soil science field, especially considering the widespread application of gold nanoparticles (AuNPs) in biological imaging. I would like a few minor comments/questions addressed before the manuscript is accepted for publication in RSOS:

1. Why were 100-nm size AuNPs chosen? Any rationale? I would expect their aggregation characteristics to depend on the particle size. What about the choice of concentration, any particular reason why the 3.84×10^{-2} mg/mL was chosen?
2. I understand and agree with the use of silica/alumina medium as worst-case scenario reference. At the same time, the use of a very simple "soil", like natural sand, with a relatively homogeneous density would have been a good addition to the 3 real soils. It could have provided a dataset where higher-density soil minerals with a high x-ray attenuation are not in the way of seeing the AuNPs, and perhaps a check on whether most of the AuNPs are actually seen in the Bangor, Dundee, and Nottingham soils. Did they calculate what percentage of the Au can actually be seen (segmented) from the XCT data?

Review form: Reviewer 2

Is the manuscript scientifically sound in its present form?

Yes

Are the interpretations and conclusions justified by the results?

Yes

Is the language acceptable?

Yes

Do you have any ethical concerns with this paper?

No

Have you any concerns about statistical analyses in this paper?

No

Recommendation?

Accept with minor revision (please list in comments)

Comments to the Author(s)

Title: Stabilising gold nanoparticles for use in X-ray computed tomography imaging of soil systems

Summary.

The authors describe how gold nanoparticles (AuNPs) can be used as a contrast agent for X-ray computed tomography (XCT) in soils. Specifically they discuss how uncoated AuNPs tend to aggregate in soil-water situations and how coating AuNPs with polyethylene glycol (PEG) can reduce aggregation such that PEG-AuNPs can flow through soil samples. The stated goal of this work was to begin developing tools and methodologies to use AuNPs as a contrast agent so that soil biology and soil physics can be studied in a non-destructive way using XRT, benefitting from the generation of 3D volume data.

The authors have succeeded in laying a solid groundwork for this research, using a methodical and detailed approach to address challenges in using AuNPs as a contrast agent in soil conditions. I have very few specific suggestions (see below) to improve on what has been presented. I recommend that this manuscript be published, but would encourage the authors to explore greater analysis of the 3D volumes presented in the syringe tubes with and without the AuNPs. Since they have already been scanned with XRT, it would be valuable to analyse the 3D volumes and ideally segment regions of differential contrast, i.e. soil, water, AuNPs. Segmentation is by no means trivial but as the 3D scan data presumably already exist, a variety of software packages can be employed to explore this area. The AuNPs have high attenuation coefficients as the authors contend, and that should allow segmentation of AuNPs from the other materials. A demonstration of that would be valuable to illustrate the central point of using AuNPs as a contrast agent in soils.

Specific suggestions.

Line 119. 'media' should be 'medium'

Line 428. Add a comma after 'PEG-AuNPs'

Table 2. In addition to the XCT scan parameters listed, it is helpful to other X-ray imaging scientists to know the source-sample-detector geometry in order to calculate X-ray dosage to the sample (critical for subsequent research with roots or other living systems) and to duplicate scan resolution on different instruments. Also, please add/include the detector specs, e.g. size, pixel pitch, resolution.

Figure 7. There's a red bounding box that (in my PDF version) overlaps the first line of text. Also, I believe the word 'which' should be 'with' in line 656. The first line of text in Figure 7 was not easy for me to follow, apologies if this is my mistake but perhaps make that more clear.

Figures S1 and S2.. Please specify if the AuNPs used were PEG-coated or not; one can discover this information in the text but it's useful to have it spelled out in the figure legend.

Decision letter (RSOS-190769.R0)

23-Aug-2019

Dear Professor Roose

On behalf of the Editors, I am pleased to inform you that your Manuscript RSOS-190769 entitled "Stabilising Gold Nanoparticles for use in X-ray Computed Tomography Imaging of Soil Systems" has been accepted for publication in Royal Society Open Science subject to minor revision in accordance with the referee suggestions. Please find the referees' comments at the end of this email.

The reviewers and handling editors have recommended publication, but also suggest some minor revisions to your manuscript. Therefore, I invite you to respond to the comments and revise your manuscript.

- Ethics statement

- Data accessibility

If you wish to submit your supporting data or code to Dryad (<http://datadryad.org/>), or modify your current submission to dryad, please use the following link:
<http://datadryad.org/submit?journalID=RSOS&manu=RSOS-190769>

- Competing interests

- Authors' contributions

- Acknowledgements

- Funding statement

Because the schedule for publication is very tight, it is a condition of publication that you submit the revised version of your manuscript before 01-Sep-2019. Please note that the revision deadline will expire at 00.00am on this date. If you do not think you will be able to meet this date please let me know immediately.

on behalf of Dr Philip Benson (Associate Editor) and Jon Blundy (Subject Editor)
openscience@royalsociety.org

Reviewer comments to Author:

Reviewer: 1

The manuscript provides evidence of stabilization of gold nanoparticles (preventing their aggregation) for application as x-ray contrast agent in soils. After showing that the nanoparticles aggregate when not coated preventing their applicability for day-long experiments, the team coated the nanoparticles with PEG, and achieved stability for a period of several days. To further confirm their applicability for soils, they used time-resolved x-ray CT combined with image-based modeling to evaluate nanoparticle diffusion in wet soil.

I think the paper would stand as an important contribution to the x-ray imaging and soil science field, especially considering the widespread application of gold nanoparticles (AuNPs) in biological imaging. I would like a few minor comments/questions addressed before the manuscript is accepted for publication in RSOS:

1. Why were 100-nm size AuNPs chosen? Any rationale? I would expect their aggregation characteristics to depend on the particle size. What about the choice of concentration, any particular reason why the 3.84×10^{-2} mg/mL was chosen?
2. I understand and agree with the use of silica/alumina medium as worst-case scenario reference. At the same time, the use of a very simple "soil", like natural sand, with a relatively homogeneous density would have been a good addition to the 3 real soils. It could have provided a dataset where higher-density soil minerals with a high x-ray attenuation are not in the way of seeing the AuNPs, and perhaps a check on whether most of the AuNPs are actually seen in the Bangor, Dundee, and Nottingham soils. Did they calculate what percentage of the Au can actually be seen (segmented) from the XCT data?

Reviewer: 2

Comments to the Author(s)

Title: Stabilising gold nanoparticles for use in X-ray computed tomography imaging of soil systems

Summary.

The authors describe how gold nanoparticles (AuNPs) can be used as a contrast agent for X-ray computed tomography (XCT) in soils. Specifically they discuss how uncoated AuNPs tend to aggregate in soil-water situations and how coating AuNPs with polyethylene glycol (PEG) can reduce aggregation such that PEG-AuNPs can flow through soil samples. The stated goal of this work was to begin developing tools and methodologies to use AuNPs as a contrast agent so that soil biology and soil physics can be studied in a non-destructive way using XRT, benefitting from the generation of 3D volume data.

The authors have succeeded in laying a solid groundwork for this research, using a methodical and detailed approach to address challenges in using AuNPs as a contrast agent in soil conditions. I have very few specific suggestions (see below) to improve on what has been presented. I recommend that this manuscript be published, but would encourage the authors to explore greater analysis of the 3D volumes presented in the syringe tubes with and without the AuNPs. Since they have already been scanned with XRT, it would be valuable to analyse the 3D volumes and ideally segment regions of differential contrast, i.e. soil, water, AuNPs. Segmentation is by no means trivial but as the 3D scan data presumably already exist, a variety of software packages can be employed to explore this area. The AuNPs have high attenuation coefficients as the authors contend, and that should allow segmentation of AuNPs from the other materials. A demonstration of that would be valuable to illustrate the central point of using AuNPs as a contrast agent in soils.

Specific suggestions.

Line 119. 'media' should be 'medium'

Line 428. Add a comma after 'PEG-AuNPs'

Table 2. In addition to the XCT scan parameters listed, it is helpful to other X-ray imaging scientists to know the source-sample-detector geometry in order to calculate X-ray dosage to the sample (critical for subsequent research with roots or other living systems) and to duplicate scan resolution on different instruments. Also, please add/include the detector specs, e.g. size, pixel pitch, resolution.

Figure 7. There's a red bounding box that (in my PDF version) overlaps the first line of text. Also, I believe the word 'which' should be 'with' in line 656. The first line of text in Figure 7 was not easy for me to follow, apologies if this is my mistake but perhaps make that more clear.

Figures S1 and S2.. Please specify if the AuNPs used were PEG-coated or not; one can discover this information in the text but it's useful to have it spelled out in the figure legend.

Author's Response to Decision Letter for (RSOS-190769.R0)

See Appendix A.

Decision letter (RSOS-190769.R1)

25-Sep-2019

Dear Professor Roose,

I am pleased to inform you that your manuscript entitled "Stabilising Gold Nanoparticles for use in X-ray Computed Tomography Imaging of Soil Systems" is now accepted for publication in Royal Society Open Science.

At this stage, we ask that you please provide the individual main figure files (publication-quality EPS or PDF format preferred) and table files (editable Word or Excel format preferred) for your submission via return email for production purposes. Please email these files as attachments to: openscience@royalsociety.org

You can then expect to receive a proof of your article in the near future. Please contact the editorial office (openscience_proofs@royalsociety.org and openscience@royalsociety.org) to let us know if you are likely to be away from e-mail contact -- if you are going to be away, please nominate a co-author (if available) to manage the proofing process, and ensure they are copied into your email to the journal.

Best regards,

on behalf of Dr Philip Benson (Associate Editor) and Jon Blundy (Subject Editor)
openscience@royalsociety.org

Follow Royal Society Publishing on Twitter: [@RSocPublishing](https://twitter.com/RSocPublishing)

Appendix A

Dear Editors and Reviewers of Royal Society Open Science,

Thank you for taking the time to review our manuscript and for the positive response to our work. We are pleased that the reviewers find the work to be novel and of interest to the research community. We believe the changes and amendments the reviewers have suggested in their comments will help to clarify the outcomes of the work. Below you will find details on the changes to the manuscript to resolve the reviewers' comments. The reviewers comments are in *italics*, text in the updated manuscript is in *inline*.

Reviewer 1:

The manuscript provides evidence of stabilization of gold nanoparticles (preventing their aggregation) for application as x-ray contrast agent in soils. After showing that the nanoparticles aggregate when not coated preventing their applicability for day-long experiments, the team coated the nanoparticles with PEG, and achieved stability for a period of several days. To further confirm their applicability for soils, they used time-resolved x-ray CT combined with image-based modelling to evaluate nanoparticle diffusion in wet soil.

I think the paper would stand as an important contribution to the x-ray imaging and soil science field, especially considering the widespread application of gold nanoparticles (AuNPs) in biological imaging. I would like a few minor comments/questions addressed before the manuscript is accepted for publication in RSOS:

Comment 1: Why were 100-nm size AuNPs chosen? Any rationale? I would expect their aggregation characteristics to depend on the particle size. What about the choice of concentration, any particular reason why the 3.84×10^{-2} mg/mL was chosen?

Both of these are very good questions. AuNPs of 100 nm in size were chosen for two main reasons. The first was that 100 nm was sufficiently large such that should we introduce plants into the system in future work then AuNPs of 100 nm are too large to actually be taken up through plant root tissues. This has now been clarified in the text of the document in Line 187:

AuNPs of 100 nm size were chosen for two reasons. Firstly, were plants to be included within experimental setups of future work, 100 nm AuNPs are too large to be taken up through pores in plant root cell walls [27].

The second reason for the use of a 100 nm particle size was that the protocol for PEG coating and concentration of the AuNPs required several rounds of centrifugation. This centrifugation is considerably easier to undertake when particle sizes are larger as they will have greater mass and as such require lower rotations per minute (RPM). This has now also been clarified in the text of the document in Line 190:

Secondly, the polymer coating procedure, described in section 2.5 Polymer Coating Process, required several rounds of centrifugation. The ease of performing centrifugation is considerably improved for larger particle sizes as they possess a greater mass and thus require a lower RPM for successful centrifugation.

With regard to the choice of concentration, there were again two main reasons for selecting the concentrations of AuNPs used for this investigation. Initial radiography contrast assessments in addition previous work from the literature had indicated that suspensions of approximately 3.84×10 mg/mL would provide sufficient contrast against soil and water (Xu *et al.*, 2008 – reference 11 in the manuscript; Sun *et al.*, 2009 – reference 10 in the manuscript). This has been addressed in line 261 of the manuscript text:

This concentration, 100 times the concentration of the suspension as supplied from the manufacturer, was used to provide contrast in XCT assessments [10,11].

Secondly, the concentration of AuNPs as supplied by Sigma Aldrich was 3.84×10^{-2} mg/mL. As such, 3.84×10 mg/mL was a concentration that would both provide sufficient contrast against soil/ water and be a simple concentration to consistently attain through a centrifugation protocol. This is because it is simply 100 times the concentration of the as-supplied AuNP suspension. We have clarified in the text in line 186 that the AuNPs were supplied from the manufacturer at 3.84×10^{-2} mg/mL, hence the use of this as the initial concentration for UV-Vis:

For the UV-Vis analysis, 100 nm AuNPs (Sigma Aldrich, UK) at the manufacturer-supplied concentration of 3.84×10^{-2} mg Au/ mL were added to aliquots of each of the three soil solutions.

Comment 2: I understand and agree with the use of silica/alumina medium as worst-case scenario reference. At the same time, the use of a very simple "soil", like natural sand, with a relatively homogeneous density would have been a good addition to the 3 real soils. It could have provided a dataset where higher-density soil minerals with a high x-ray attenuation are not in the way of seeing the AuNPs, and perhaps a check on whether most of the AuNPs are actually seen in the Bangor, Dundee, and Nottingham soils. Did they calculate what percentage of the Au can actually be seen (segmented) from the XCT data?

All of the above are very well made points. We did not use a sand material within this investigation for a few reasons. The three soils that we have used are more chemically complex than sand with regard to the major factors affecting AuNP stability i.e. surface charge, acidity and salinity. We therefore assume that if the AuNPs are able to remain stable in the complex chemical environments of the three soils then they will also remain stable within sand. Experiments using sand could form a worthwhile future experiment and we have now commented on this in line 513 in the discussion:

Given the complexity of the chemical environment provided by soil and the variability in soil mineral attenuation it would likely be worthwhile to undertake future investigations initially using sand. This would provide a less complex chemical environment and the X-ray attenuation of sand mineral grains would likely be more homogeneous.

With regard to the possibility of segmenting the AuNPs in the XCT data, we have now added an additional figure (Figure 8) to the manuscript which demonstrates the possibility for segmenting contrast media from soil and also from silica/ alumina. This figure is referred to in lines 403, 436, 493 and 540 of the manuscript as follows:

Line 403

These voxels were of such high grey value that they can be segmented from silica and alumina using a simple grey value threshold (**Figure 8**).

Line 436

The contrast between PEG-AuNPs and soil means that the PEG-AuNP suspension can also be segmented using only a simple grey value threshold (**Figure 8**).

Line 493

These regions provided considerable contrast (**Figures S1 & S2**) and also meant the PEG-AuNPs could be segmented from the silica/ alumina using a grey value threshold (**Figure 8**).

Line 540

The contrast between the PEG-AuNPs and the soil particles also meant that the PEG-AuNPs can be easily segmented using only a simple grey value threshold (**Figure 8**).

Quantifying the amount AuNPs that are visible and stable within the soil using XCT is difficult using the data we have available. However, relative quantification is possible using UV-Vis and we have demonstrated using UV-Vis that nearly 100% of PEG-AuNPs remained stable within the three soils. This is made evident by the near-identical height of the peaks in Figure 6b between wavelengths of 572 and 580 nm. This is referred to in line 528 of the manuscript text:

This was indicated by the presence of the characteristic absorption peak between the wavelengths of 572 and 580. In particular, the absorption values for the characteristic AuNP peaks in Figure 6b demonstrate this AuNP stability. This is because the concentration of AuNPs extracted from the base of the columns was almost equal to that of the AuNP suspensions applied to the soil surface in the column three days earlier.

Reviewer 2:

The authors have succeeded in laying a solid groundwork for this research, using a methodical and detailed approach to address challenges in using AuNPs as a contrast agent in soil conditions. I have very few specific suggestions (see below) to improve on what has been presented. I recommend that this manuscript be published, but would encourage the authors to explore greater analysis of the 3D volumes presented in the syringe tubes with and without the AuNPs.

Comment 1: Since they have already been scanned with XRT, it would be valuable to analyse the 3D volumes and ideally segment regions of differential contrast, i.e. soil, water, AuNPs. Segmentation is by no means trivial but as the 3D scan data presumably already exist, a variety of software packages can be employed to explore this area. The AuNPs have high attenuation coefficients as the authors contend, and that should allow segmentation of AuNPs from the other materials. A demonstration of that would be valuable to illustrate the central point of using AuNPs as a contrast agent in soils.

We have now added an additional figure (Figure 8) to the manuscript which demonstrates the possibility for segmenting the AuNP contrast media from soil and also from silica/ alumina. This figure is referred to in lines 403, 436, 493 and 540 of the manuscript as follows:

Line 403

These voxels were of such high grey value that they can be segmented from silica and alumina using a simple grey value threshold (**Figure 8**).

Line 436

The contrast between PEG-AuNPs and soil means that the PEG-AuNP suspension can also be segmented using only a simple grey value threshold (**Figure 8**).

Line 493

These regions provided considerable contrast (**Figures S1 & S2**) and also meant the PEG-AuNPs could be segmented from the silica/ alumina using a grey value threshold (**Figure 8**).

Line 540

The contrast between the PEG-AuNPs and the soil particles also meant that the PEG-AuNPs can be easily segmented using only a simple grey value threshold (**Figure 8**).

Comment 2:

Line 119. 'media' should be 'medium'

Line 428. Add a comma after 'PEG-AuNPs'

Both of these changes have now been made to the manuscript. Thank you for noticing these and providing the amendments.

Comment 3: Table 2. In addition to the XCT scan parameters listed, it is helpful to other X-ray imaging scientists to know the source-sample-detector geometry in order to calculate X-ray dosage to the sample (critical for subsequent research with roots or other living systems) and to duplicate scan resolution on different instruments. Also, please add/include the detector specs, e.g. size, pixel pitch, resolution.

Table 2 has now been expanded to include the following further imaging parameters: detector dimensions (pixels²), pixel size on detector (μm^2), detector dimension (cm²), source to sample distance (mm), source to detector distance (mm).

Comment 4: Figure 7. There's a red bounding box that (in my PDF version) overlaps the first line of text. Also, I believe the word 'which' should be 'with' in line 656. The first line of text in Figure 7 was not easy for me to follow, apologies if this is my mistake but perhaps make that more clear.

We have reformatted Figure 7 such that the issue with the red bounding box should be resolved, and the first line of Figure 7 has now been rewritten as follows and we hope this sentence now provides more clarity:

Figure 7. The XCT contrast provided by a suspension of PEG-AuNPs within a syringe barrel containing soil particles.

Comment 5: Figures S1 and S2.. Please specify if the AuNPs used were PEG-coated or not; one can discover this information in the text but it's useful to have it spelled out in the figure legend.

We have now amended the figure captions for Figures S1 and S2 to specify the coating applied to the AuNPs in these instances and believe these figures are now more standalone. The captions have been rewritten as follows to improve clarity:

Supplementary Figure S1. Horizontal image slices from XCT images of silica and/ or alumina displaying the accumulation of PEG-AuNPs on the silica and alumina surfaces.

Supplementary Figure S2. XCT images of silica and/ or alumina confirming the accumulation of PEG-AuNPs on the very surface of the silica and alumina in the columns.